

# Increased CD14$^+$HLA-DR$^{-/\text{low}}$ myeloid-derived suppressor cells can be regarded as a biomarker on disease severity and response to therapy in acute coronary syndrome

Yinsheng Tan[1,2,*], Mingde Ren[3,*], Jie Hou[2], Tao Hou[2] and Xianhe Lin[1]

[1] Department of Cardiovascular, First Affiliated Hospital of Anhui Medical University, Hefei, Anhui Province, China

[2] Department of Cardiovascular, East District of First Affiliated Hospital of Anhui Medical University, Hefei, Anhui Province, China

[3] Department of Cardiovascular, Lu'an People's Hospital of Anhui Province, Lu'an, Anhui Province, China

[*] These authors contributed equally to this work.

Corresponding author
Xianhe Lin,
changyu8674004@163.com

## ABSTRACT

**Purpose.** This study aimed to investigate the dynamic changes in monocytic myeloid-derived suppressor cells (M-MDSCs) and their implications in the pathogenesis of acute coronary syndrome (ACS), shedding light on potential therapeutic targets.

**Experimental Design.** Peripheral blood samples were collected from 68 ACS patients, 35 stable angina pectoris (SAP) patients, and 30 healthy controls (HC). Multi-parameter flow cytometry was employed for analysis of M-MDSCs, explored with disease characteristics and progression.

**Results.** ACS patients exhibited an increased frequency of circulating M-MDSCs compared to SAP patients and HC. M-MDSCs levels demonstrated associations with ACS type, coronary artery lesions, multi-vessel disease, and cardiac dysfunction severity. Higher M-MDSCs levels were found in obese patients. Notably, therapy led to a significant decrease in M-MDSCs frequency. Furthermore, ACS patients exhibited elevated levels of interleukin (IL)-6, IL-10, granulocyte-macrophage colony-stimulating factor (GM-CSF), and tumor necrosis factor-$\alpha$ (TNF-$\alpha$) in the cytokine profile associated with M-MDSCs. Increased expression of arginase-1(Arg-1) was observed in ACS patients, with positive correlations between M-MDSCs levels and IL-6, GM-CSF, and Arg-1 expression. The diagnostic performance of triglycerides (TG), high-density lipoprotein cholesterol (HDL-C), and M-MDSCs levels varied in predicting the severity of coronary artery stenosis, with TG showing higher specificity, HDL-C displaying higher sensitivity, and M-MDSCs levels demonstrating balanced sensitivity and specificity.

**Conclusions.** Assessment of M-MDSCs frequency holds promise as a predictive marker for disease progression and therapy response of coronary artery stenosis. The elevated presence of M-MDSCs suggests their potential role in modulating ACS-related inflammation.

**How to cite this article** Tan Y, Ren M, Hou J, Hou T, Lin X. 2024. Increased CD14$^+$HLA-DR$^{-/\text{low}}$ myeloid-derived suppressor cells can be regarded as a biomarker on disease severity and response to therapy in acute coronary syndrome. *PeerJ* **12**:e18154
http://doi.org/10.7717/peerj.18154

## INTRODUCTION

Acute coronary syndrome (ACS), encompassing unstable angina and myocardial infarction, is a leading cause of morbidity and mortality worldwide, posing a significant burden on healthcare systems (*Nohria & Antono, 2024*). Despite advancements in medical interventions and revascularization strategies, the pathophysiological mechanisms underlying ACS development and progression remain complex and multifactorial. In recent years, there has been growing recognition of the pivotal role played by immune cells in atherosclerosis and ACS, highlighting the need for a comprehensive understanding of the immunopathogenesis underlying this disorder (*Delaney et al., 2021*; *Riesinger et al., 2018*). Pathologically, chronic inflammation of the arterial wall, characterized by immune activation and the release of leukocytes (*e.g.*, monocytes, neutrophils, and lymphocytes) into the susceptible regions of the intima, is a hallmark of arteriosclerosis (*Askari et al., 2021*; *Shi et al., 2023*). Immune activation is considered the most critical feature determining the vulnerability of atherosclerotic plaques (*Morici et al., 2020*). Understanding the intricate interplay between various cellular components of the immune system and the vascular endothelium is crucial for unraveling the pathogenesis of ACS and identifying potential therapeutic targets.

Myeloid-derived suppressor cells (MDSCs) have emerged as intriguing players with potential implications for disease pathogenesis and clinical management (*Zhang et al., 2024*). MDSCs constitute a heterogeneous population of immature myeloid cells characterized by their potent immunosuppressive properties (*Zhou et al., 2023*). There are two main subtypes of MDSCs: granulocytic MDSCs (G-MDSCs) and monocytic MDSCs (M-MDSCs), distinguished by their phenotypic markers. In mice, MDSCs are characterized by the expression of Gr-1 and CD11b molecules (*Tian et al., 2023*). In humans, M-MDSCs are commonly identified as $CD11b^+CD33^+CD14^+HLA\text{-}DR^{low/-}$ or $CD14^+HLA\text{-}DR^{low/-}$ cells, while G-MDSCs are typically characterized as $CD11b^+CD33^+CD15^+HLA\text{-}DR^+$ cells (*Ostrand-Rosenberg, Lamb & Pawelec, 2023*). MDSCs employ various mechanisms to suppress immune responses, including direct cell–cell contact and the secretion of immunomodulatory factors (*Mizuhara et al., 2023*; *Zahran et al., 2020*). These cells play a pivotal role in maintaining immune homeostasis, regulating inflammatory responses, and modulating the balance between innate and adaptive immunity. Importantly, aberrant expansion and activation of MDSCs have been implicated in the pathogenesis of various pathological conditions, including cancer, infections, and autoimmune diseases (*Gjerstorff, 2023*; *Li et al., 2023*; *Li, Ahirwar & Wu, 2023*; *Wang et al., 2019*).

Despite the well-established role of MDSCs in various pathological conditions, their involvement in cardiovascular diseases, particularly ACS, remains relatively unexplored. Recent studies have suggested that MDSCs may play a significant role in shaping the inflammatory milieu within atherosclerotic plaques, thereby influencing disease
progression (*Li, Li & Jia, 2023*; *Xing et al., 2020*). However, comprehensive investigations into the frequency, phenotype, and functional characteristics of MDSCs in ACS patients are lacking. In-depth research on the immunosuppressive and pro-inflammatory functions of MDSCs in ACS patients, as well as their interactions with other immune cells, will contribute to a more comprehensive understanding of the role of MDSCs in the pathogenic mechanisms of ACS. These research findings not only will enrich our knowledge of the role of MDSCs in cardiovascular diseases but also may provide new targets and strategies for the development of immune-based therapeutic approaches for ACS.

This study aims to investigate the frequency of circulating M-MDSCs in ACS patients and explore their potential association with clinical characteristics.

## MATERIALS AND METHODS

### Human samples

Fresh paired peripheral blood (PB) samples were collected from 68 ACS patients and 35 SAP patients at the First Affiliated Hospital of Anhui Medical University and Feidong People's Hospital between January 2023 and January 2024. Specifically, the diagnosis of ACS and SAP refers to the clinical diagnostic criteria of ischemic heart disease of the International Society of Cardiology and the World Health Organization (ISFC/WHO) (*Braunwald et al., 2000*). Additionally, PB samples from 30 healthy donors were included as the healthy control (HC) group. The basic characteristics of the ACS, SAP, and HC participants are summarized in Table 1. The degree of coronary artery stenosis was assessed using the Gensini scoring system (*Rampidis et al., 2019*), where a total score of ≤24 indicated mild stenosis, 25–49 indicated moderate stenosis, and ≥50 indicated severe stenosis. All patients underwent percutaneous coronary intervention (PCI) and received dual antiplatelet therapy with aspirin and clopidogrel. This study was conducted in accordance with the Declaration of Helsinki, and the study protocol was approved by the Ethics Committee of the First Affiliated Hospital of Anhui Medical University (*NO. S20200021*). Written informed consent was obtained from all patients and healthy donors, and samples were processed within six hours of collection. Portions of this text were previously published as part of a preprint (https://doi.org/10.21203/rs.3.rs-4461420/v1).

### Sample collection

Fresh PB samples were collected from all cohorts using heparin-containing tubes. For the ACS patients, paired pre- and post-therapy PB samples were collected to assess the impact of standard PCI treatment or anticoagulant therapy on the study parameters. All PB samples were collected prior to any treatments, ensuring that the measurements captured the baseline levels for accurate analysis.

### Surface staining

PB samples (100 μL) were collected and transferred into test tubes. To facilitate staining, the appropriate concentration of the following fluorescently labeled monoclonal antibodies was added: Human TruStain FcX TM (Clone 422301; BioLegend, San Diego, CA, USA), Brilliant Violet 510 TM anti-human CD45 (Clone 2D1; BioLegend), PE anti-human

**Table 1  Clinical characteristics for ACS, SAP patients and HC.**

| Characteristic | ACS | SAP | HCs | F/$\chi^2$ | P | ACS vs HCs | | SAP vs HCs | | ACS vs SAP | |
|---|---|---|---|---|---|---|---|---|---|---|---|
| Case | 68 | 35 | 30 | | | F/$\chi^2$ | P | F/$\chi^2$ | P | F/$\chi^2$ | P |
| Sex (male/female) | 38/30 | 20/15 | 18/12 | 0.144 | 0.930 | | | | | | |
| Age (years) | 55.1 ± 9.5 | 53.6 ± 8.9 | 54.5 ± 9.3 | 0.121 | 0.888 | | | | | | |
| Hypertension (%) | 41(60.2%) | 23(65.7%) | 8(26.7%) | 12.045 | 0.002 | 9.416 | 0.002 | 9.873 | 0.002 | 0.289 | 0.591 |
| Smoking (%) | 14(20.6%) | 8(22.8%) | 6(20%) | 0.097 | 0.952 | | | | | | |
| Obesity (%) | 20(29.4%) | 11(31.4%) | 8(26.7%) | 0.177 | 0.915 | | | | | | |
| Diabetes (%) | 22(32.4%) | 9(25.7%) | 3(10%) | 6.186 | 0.045 | 5.473 | 0.019 | 2.65 | 0.104 | 0.484 | 0.487 |
| TG(mmol/L) | 1.73 ± 0.81 | 1.58 ± 0.46 | 1.21 ± 0.42 | 17.231 | 0.003 | 3.719 | 0.001 | 1.2 | 0.621 | 1.013 | 0.313 |
| TC(mmol/L) | 4.71 ± 1.53 | 4.56 ± 1.33 | 4.19 ± 0.78 | 2.338 | 0.177 | | | | | | |
| HDL-C(mmol/L) | 0.96 ± 0.28 | 1.29 ± 0.33 | 1.43 ± 0.23 | 28.063 | 0.001 | 7.549 | 0.001 | 1.981 | 0.09 | 5.327 | 0.001 |
| LDL-C(mmol/L) | 2.88 ± 0.83 | 2.49 ± 0.77 | 1.32 ± 0.32 | 59.017 | 0.001 | 9.949 | 0.001 | 6.647 | 0.001 | 2.314 | 0.023 |

HLA-DR (Clone LN3; BioLegend), FITC anti-human CD14 (Clone M5E2). The samples were then incubated for 15 min at room temperature to allow for antibody binding and cell surface staining.

Subsequently, red blood cell lysis was performed using red blood cell lysis buffer from BD Biosciences. The lysis process lasted for 15 min, ensuring the removal of red blood cells while preserving the stained cells of interest. After completion, the stained cells were washed twice with phosphate-buffered saline (PBS) to eliminate any remaining debris or unbound antibodies.

Data acquisition and analysis were carried out using a flow cytometer (Cytoflex S; Beckman Coulter, Brea, CA, USA) and the accompanying CytExpert software (Beckman Coulter). M-MDSCs were characterized by the expression of $CD14^+HLA-DR^{-/low}$ on $CD14^+$ monocytes in this study.

## Cytokine assay

Simultaneously with PB collection, plasma samples were obtained from all individuals. The PB samples were centrifuged at 3,000 rpm for 10 min at 4 °C to separate the plasma fractions. The obtained plasma fractions were carefully collected and stored at −80 °C until further use to maintain sample integrity. To assess the concentrations of IL-6, IL-10, GM-CSF, and TNF-$\alpha$ in the plasma samples, ELISA Kits (ESM; R&D System) were employed following the manufacturer's instructions.

## RT-PCR analysis

Total RNA was extracted from Peripheral blood mononuclear cells (PBMCs) using TRizol reagent (Invitrogen, Carlsbad, CA, USA). The extracted RNA was then subjected to reverse transcription using the First-Strand Synthesis System (TaKaRa, Dalian, China) to generate first-strand cDNA. Real-time PCR analysis was performed using the Applied Biosystems 7500 Real-time Polymerase Chain Reaction system. SYBR Green PCR Master mix was used in the PCR reactions. The mRNA levels of Arg-1 and inducible nitric oxide synthase (iNOS) were analyzed in each reaction. The following primer

sequences were used: Arg-1: forward 5′-GGCTGGTCTGCTTGAGAAAC-3′ and reverse 5′-ATTGCCAAACTGTGGTCTCC-3′. iNOS: forward 5′-CTTTCCAAGACACACTTCACCA-3′ and reverse 5′-TATCTCCTTTGTTACCGCTTCC-3′. To normalize the expression levels, glyceraldehyde phosphate dehydrogenase (GAPDH) was used as a control. The primer sequences for GAPDH were: forward 5′-CAGGAGGCCATTGCTGATGAT-3′ and reverse 5′-GAAGGCTGGGGCTCATTT-3′. The thermal cycling conditions were as follows: an initial denaturation step at 95 °C for 30 s, followed by 40 cycles of amplification with the following profile: 95 °C for 5 s and 60°C for 34 s. The relative expression levels of the target genes were determined using the following formula: $2^{-(\text{Cttarget}-\text{Ctcontrol})}$, where Ct represents the threshold cycle.

## Statistical analysis

The statistical analysis of the data was performed using GraphPad Prism 8 and SPSS 27.0 software. The experimental data were presented as mean ± standard deviation (SD). For comparisons among multiple groups, analysis of variance (ANOVA) was conducted, followed by *post hoc* pairwise comparisons using the Least Significant Difference (LSD) $t$-test. When comparing two paired samples, either the non-parametric unpaired Mann–Whitney U-test or the parametric Student's $t$-test was employed. To assess correlations between M-MDSCs frequency and each parameter, the Pearson's coefficient test was utilized. Logistic regression analysis was employed to identify factors influencing the occurrence of middle to serious coronary artery disease. Receiver operating characteristic (ROC) curves were constructed to determine the optimal cut-off value based on the maximum Youden index. In all statistical tests, a *p*-value less than 0.05 (*P < 0.05*) was considered statistically significant.

## RESULTS

### Comparative clinical analysis of ACS, SAP and control group

The clinical data of the three study groups, including initial diagnosis ACS patients, SAP patients, and the HC group, are summarized in Table 1. No significant differences were observed in gender, age, smoking rate, obesity incidence, and total cholesterol (TC) levels among the three groups ($P > 0.05$).

However, the prevalence of hypertension (60.2%) and diabetes (32.4%) was significantly higher in the ACS group compared to the SAP group (65.7% and 25.7%) and the HC group (26.7% and 10%) ($P < 0.05$). Additionally, the ACS group had higher TG levels (1.73 ± 0.81 mmol/L) compared to the SAP group (1.58 ± 0.46 mmol/L) and the HC group (1.21 ± 0.42 mmol/L) ($P < 0.01$), while the HDL-C level (0.96 ± 0.28 mmol/L) was lower than the other two groups ($P < 0.01$). The LDL-C level showed significant differences among the three groups, with the ACS group having the highest level (2.88 ± 0.83 mmol/L), followed by the SAP group (2.49 ± 0.77 mmol/L), and the HC group having the lowest level (1.32 ± 0.32 mmol/L) ($P < 0.01$).

### Increased frequency of circulating M-MDSCs in ACS patients

In this study, the frequency of circulating M-MDSCs was investigated in ACS patients. Flow cytometry analysis was performed, and M-MDSCs were characterized by the expression

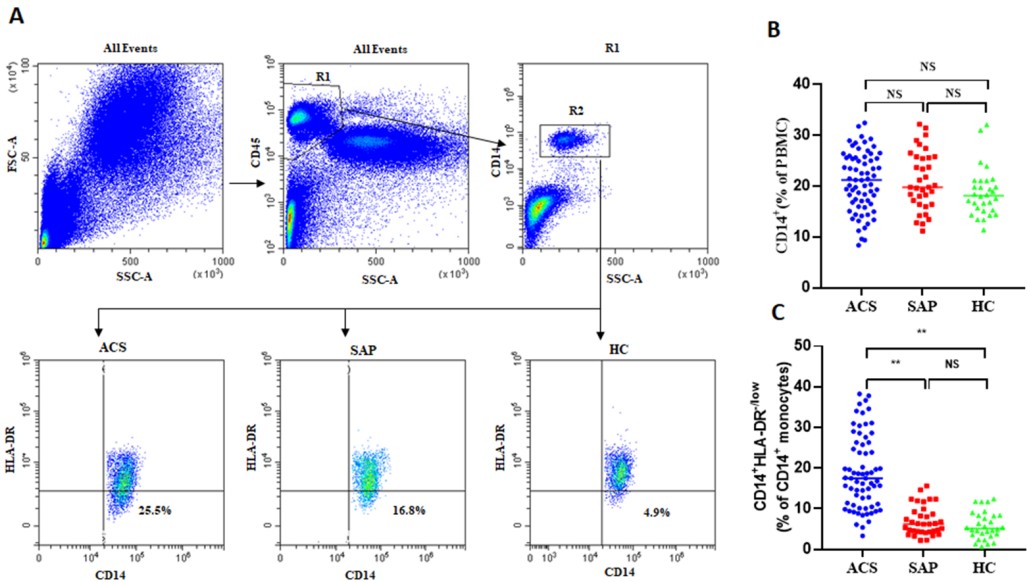

**Figure 1** **The expression of M-MDSCs in ACS patients, SAP patients and HC.** (A) Flow cytometry data plots illustrating the gating strategy of human peripheral blood M-MDSCs. (B) No significant differences were observed in the frequency of CD14$^+$ monocytes (in PBMCs) between ACS patients, SAP patients, and HC. (C) The levels of M-MDSCs (in CD14$^+$ monocytes) were significantly increased in ACS patients and SAP patients compared to HC. Each point represents an individual. The horizontal bar represents the average. **$P < 0.01$; *ns*, non-significant.

of CD14$^+$HLA-DR$^{-/low}$ on CD14$^+$ monocytes. Flow cytometry data plots illustrating the gating strategy of human peripheral blood M-MDSCs (Fig. 1A). No significant differences were observed in the frequency of CD14$^+$ monocytes between ACS patients, SAP patients, and HC (20.9 ± 5.82% *vs* 21.09 ± 5.72% *vs* 19.69 ± 4.69%, $P = 0.146$, Fig. 1B). However, the levels of M-MDSCs were significantly increased in ACS patients and SAP patients compared to healthy controls (18.92 ± 9.04% *vs* 7.01 ± 3.6% *vs* 5.82 ± 3.39%, $P = 0.001$, Figs. 1A, 1C).

## Association between characteristics and disease status of ACS patients with M-MDSCs

To determine the clinical significance of increased M-MDSCs in ACS patients, the patients were categorized based on different clinical characteristics. Among the total 68 ACS patients, there were 19 patients with acute ST-segment elevation myocardial infarction (STEMI), 23 patients with acute non-ST-segment elevation myocardial infarction (NSTEMI), and 26 patients with unstable angina pectoris (UA). Initially, STEMI patients exhibited a significantly higher frequency of M-MDSCs compared to NSTEMI patients and UA patients (25.28 ± 8.97% *vs* 18.36 ± 9.56% *vs* 14.76 ± 5.6%, $P = 0.001$, Fig. 2A). However, there was no significant difference in the frequency of M-MDSCs between NSTEMI patients and UA patients ($P > 0.05$). Based on the severity of coronary artery stenosis, the patients were further classified as mild (25 patients), moderate (28 patients), and severe (15 patients). Significantly elevated M-MDSCs frequency were observed in severe patients

and moderate patients compared to mild patients. (25.22 ± 8.17% *vs* 20.45 ± 9.61% *vs* 13.42 ± 5.11%, $P = 0.001$, Fig. 2B). However, no significant difference was found in the frequency of M-MDSCs between severe and moderate patients ($P > 0.05$). Considering the number of coronary artery lesions, the patients were grouped as follows: single vessel lesions (23 cases), two vessel lesions (33 cases), and three vessel lesions (12 cases). Patients with three vessel lesions had a significantly higher frequency of M-MDSCs compared to those with single vessel lesions (25.46 ± 8.3% *vs* 19.38 ± 9.05% *vs* 14.85 ± 7.37%, $P = 0.003$, Fig. 2C). However, there were no significant differences in the frequency of M-MDSCs between patients with single and two vessel lesions, as well as between patients with two and three vessel lesions ($P > 0.05$). According to the Killip classification of cardiac function, the patients were categorized into three groups: level II (24 patients), level III (31 patients), and level IV (13 patients). Patients with level IV cardiac function had a significantly higher frequency of M-MDSCs compared to those with level II and III (29.03 ± 6.88% *vs* 17.67 ± 8.17% *vs* 15.05 ± 7.11%, $P = 0.001$, Fig. 2D). However, there was no significant difference in the frequency of M-MDSCs between patients with level I and II ($P > 0.05$). Interestingly, obese patients (22 patients) had significantly higher levels of M-MDSCs compared to non-obese patients (46 patients) (23.38 ± 9.39% *vs* 16.78 ± 8.13%, $P = 0.004$, Fig. 2E). To evaluate the relationship between the frequency of M-MDSCs and treatment outcomes, all ACS patients received emergency PCI surgery and anticoagulant therapy such as aspirin and clopidogrel for more than 1 month. The frequency of M-MDSCs significantly decreased after therapy (18.92 ± 9.04% *vs* 7.79 ± 4.34%, $P = 0.001$, Fig. 2F). No significant differences were observed in the frequency of M-MDSCs among different age groups, genders, hypertension, smoking, or diabetes (Fig. S1).

## M-MDSCs associated cytokine profile in ACS patients

This study investigated the cytokine profile associated with M-MDSCs in ACS patients. The levels of IL-6, IL-10, GM-CSF, and TNF-$\alpha$ were measured in the peripheral blood of ACS and SAP patients. The results showed significantly higher plasma concentrations of IL-6, IL-10, GM-CSF, and TNF-$\alpha$ in ACS patients compared to SAP patients and HC. The plasma concentrations of IL-6 were significantly increased in ACS patients compared to SAP patients and HC (28.41 ± 9.46 pg/ml *vs* 14.5 ± 3.24 pg/ml *vs* 11.07 ± 4.1 pg/ml, $P = 0.001$, Fig. 3A). Similar findings were observed for IL-10 (56.12 ± 14.31 ng/ml *vs* 36.08 ± 13.61 ng/ml *vs* 10.56 ± 3.08 ng/ml, $P = 0.001$, Fig. 3B), GM-CSF (123 ± 34.06 pg/ml *vs* 68.3 ± 23.96 pg/ml *vs* 28.6 ± 12.89 pg/ml, $P = 0.001$, Fig. 3C), and TNF-$\alpha$ (52.6 ± 15.19 pg/ml *vs* 28.2 ± 5.97 pg/ml *vs* 11.12 ± 4.06 pg/ml, $P = 0.001$, Fig. 3D).

M-MDSCs are known to secrete various cytokines and enzymes to exert immuno-suppressive effects, including Arg-1 and iNOS. This study also measured the levels of MDSC-associated enzymes in ACS patients. It was found that the level of Arg-1 mRNA was significantly increased in ACS patients compared to SAP patients and HC (5.21 ± 1.03 *vs* 3.47 ± 1.01 *vs* 3.25 ± 1.24, $P = 0.001$, Fig. 3E). The level of iNOS mRNA was also higher in ACS patients, although not significantly (Fig. 3F).

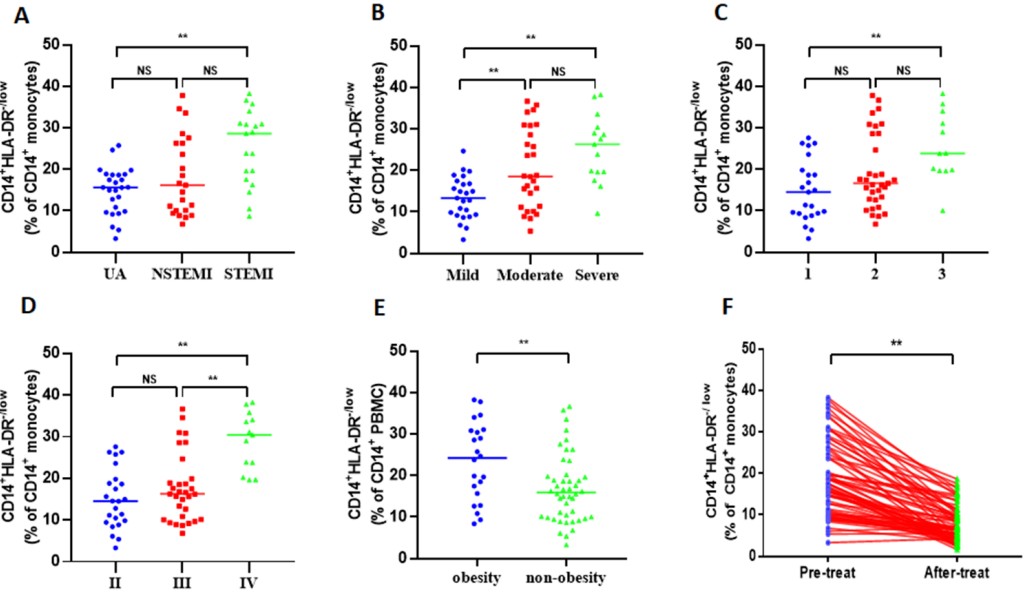

**Figure 2 Clinical relevance of M-MDSCs in ACS patients.** (A) STEMI patients exhibited a significantly higher frequency of M-MDSCs compared to NSTEMI patients and UA patients. (B) Significantly elevated M-MDSC frequency were observed in severe patients and moderate patients compared to mild patients. (C) Patients with three vessel lesions had a significantly higher frequency of M-MDSCs compared to those with single vessel lesions. (D) Patients with level IV cardiac function had a significantly higher frequency of M-MDSCs compared to those with level II and III. (E) Obese patients had significantly higher levels of M-MDSCs compared to non-obese patients. (F) The frequency of M-MDSCs significantly decreased after therapy. Each dot represents one individual. Horizontal bars indicate mean values. **$P < 0.01$; *ns*, non-significant.

Furthermore, the study analyzed the correlation between M-MDSCs levels and associated cytokines in ACS patients. Positive correlations were observed between M-MDSCs levels and the expression of IL-6 ($r = 0.64$, $P = 0.01$, Fig. 4A), GM-CSF ($r = 0.61$, $P = 0.01$, Fig. 4B), and Arg-1 ($r = 0.56$, $P = 0.01$, Fig. 4C) in ACS patients. However, no significant correlations were found between M-MDSCs levels and the expression levels of IL-10, TNF-$\alpha$, and iNOS ($P > 0.05$).

## Analysis of multiple risk factors and the degree coronary artery stenosis in ACS patients

In the logistic regression analysis, this study investigated multiple risk factors and their association with the degree of coronary artery stenosis in ACS patients. The dependent variable was the degree of stenosis, while the independent variables included sex, gender, age, hypertension, smoking, obesity, diabetes, TG, TC, HDL-C, LDL-C, and M-MDSCs. The results of the analysis, as shown in Table 2, indicated that several factors were statistically significant predictors of the severity of coronary artery stenosis in ACS patients. Specifically, diabetes, TG, HDL-C, and M-MDSCs were identified as significant risk factors. Among these factors, TG emerged as the most significant risk factor, with the highest odds ratio (OR), indicating a strong association with the degree of stenosis. Diabetes and M-MDSCs

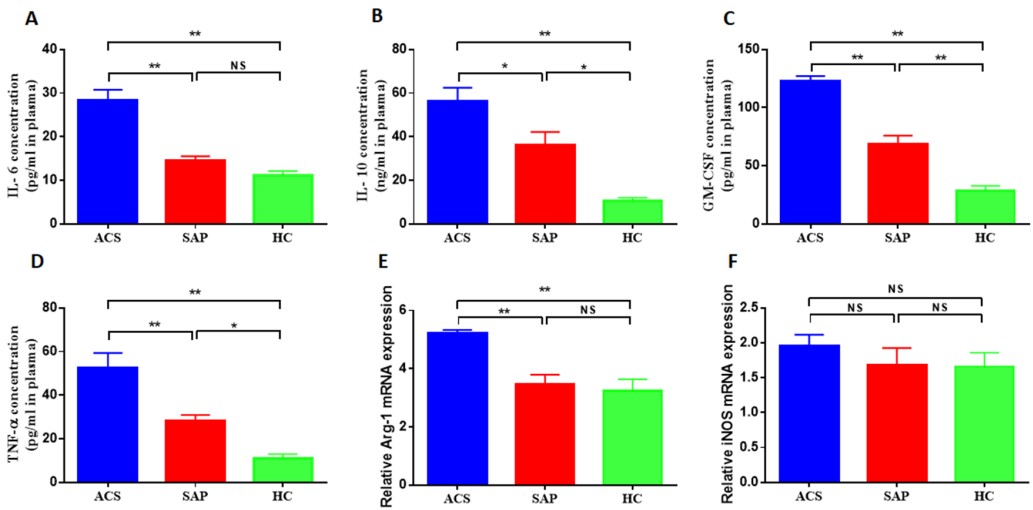

**Figure 3** **M-MDSCs associated cytokine profile in ACS patients.** (A) The plasma concentrations of IL-6 were significantly increased in ACS patients compared to SAP patients and HC. (B) The plasma concentrations of IL-10 were significantly increased in ACS patients compared to SAP patients and HC. (C) The plasma concentrations of GM-CSF were significantly increased in ACS patients compared to SAP patients and HC. (D) The plasma concentrations of TNF-$\alpha$ were significantly increased in ACS patients compared to SAP patients and HC. (E) The level of Arg-1 mRNA was significantly increased in ACS patients compared to SAP patients and HC. (F) The level of iNOS mRNA was not significantly higher in ACS patients compared to SAP patients and HC. $*P < 0.05$; $**P < 0.01$; ns, non-significant.

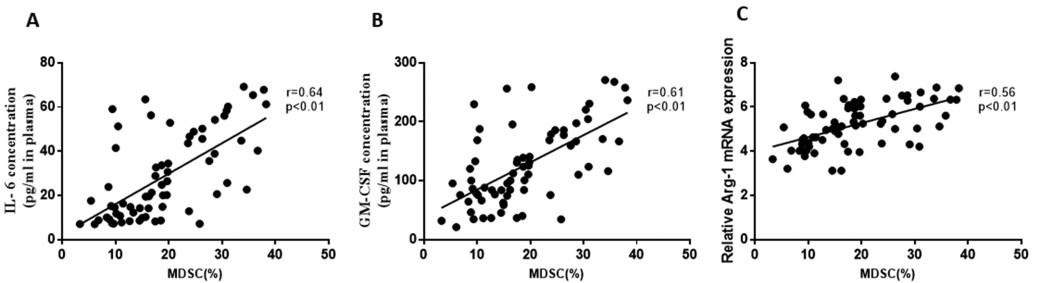

**Figure 4** **Correlation between the levels of M-MDSCs and associated cytokines in ACS patients.** (A) Positive correlations were observed between the levels of M-MDSCs and the expression of IL-6. (B) Positive correlations were observed between the levels of M-MDSCs and the expression of GM-CSF. (C) Positive correlations were observed between the levels of M-MDSCs and the expression of Arg-1 mRNA.

also showed significant associations with the severity of stenosis. Interestingly, HDL-C exhibited a significant inverse association with the degree of coronary artery stenosis.

## Multiple risk factors in predicting the severity of coronary artery stenosis

The researchers performed ROC curve analysis to assess the diagnostic performance of TG, HDL-C, and M-MDSCs levels in predicting the severity of coronary artery stenosis. The ROC curve analysis provides information on the sensitivity and specificity of these factors

**Table 2 Logistic regression analysis on the middle to serious coronary artery disease.**

| Characteristic | β | SE | Wald | P | OR(95%CI) |
|---|---|---|---|---|---|
| Diabetes | 1.59 | 0.63 | 6.45 | 0.011 | 4.91(1.44–16.77) |
| TG | 4.65 | 2.19 | 4.53 | 0.033 | 14.51(1.65–76.67) |
| HDL-C | −9.49 | 3.55 | 7.16 | 0.014 | 0.12(0.01–0.08) |
| M-MDSCs | 0.32 | 0.15 | 4.61 | 0.032 | 2.38(1.03–6.86) |

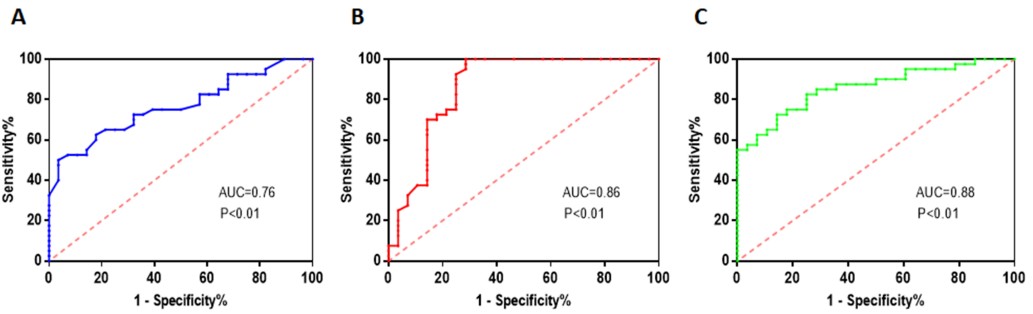

**Figure 5 Multiple risk factors in predicting the severity of coronary artery stenosis.** (A) TG has moderate diagnostic performance in predicting the severity of coronary artery stenosis. (B) HDL-C has a high ability to correctly identify patients with severe stenosis. (C) M-MDSCs has a moderate ability to correctly identify patients with severe stenosis.

at different threshold values. For TG, the area under the curve (AUC) value was found to be 0.76. The critical threshold value for TG was determined to be 1.87 mmol/L. At this threshold, the sensitivity was 50% and the specificity was 96.43%. These values indicate that TG has moderate diagnostic performance in predicting the severity of coronary artery stenosis (Fig. 5A). Regarding HDL-C, the AUC value was higher at 0.86. The critical threshold value for HDL-C was determined to be 1.18 mmol/L. At this threshold, the sensitivity was found to be 97.5%, indicating a high ability to correctly identify patients with severe stenosis. However, the specificity was 71.43%, suggesting a relatively lower ability to accurately identify patients without severe stenosis (Fig. 5B). For M-MDSCs levels, the AUC value was 0.88, indicating a good diagnostic performance. The critical threshold value for M-MDSCs was determined to be 17.55%. At this threshold, the sensitivity was 72.5%, indicating a moderate ability to correctly identify patients with severe stenosis. The specificity was higher at 85.71%, suggesting a good ability to accurately identify patients without severe stenosis (Fig. 5C).

## DISCUSSION

Several previous studies have investigated the role of MDSCs in various disease conditions, including cancer, chronic infections, and autoimmune disorders. In the field of cardiovascular diseases, limited research has been conducted on the involvement of MDSCs, particularly M-MDSCs, in ACS.

This study primarily focuses on M-MDSCs due to their higher immunosuppressive capacity compared to G-MDSCs in the tumor microenvironment, as evidenced by previous research (*Haverkamp et al., 2014*). Furthermore, it has been observed that G-MDSCs are present in whole blood but absent from PBMC, leading to relatively less research attention on G-MDSCs (*Azzaoui et al., 2016*). Therefore, our study specifically emphasizes the investigation of M-MDSCs.

In previous study, *Wang et al. (2015)* found the levels of MDSCs were significantly elevated in ACS patients compared to healthy controls. Within 24 h after acute myocardial infarction (AMI), there was a noticeable rise in the levels of MDSCs in the peripheral blood. This increase occurred in the presence of advanced glycation end-products (AGEs), indicating that the augmented infiltration of MDSCs by AGEs plays a role in exacerbating myocardial injury during AMI (*Yao et al., 2015*). Consistent with findings in previous researches, our study revealed a significant increase in the frequency of M-MDSCs in ACS patients compared to SAP patients and healthy controls. This finding the involvement of MDSCs in immune regulation and suppression under various pathological conditions. The expansion of M-MDSCs in ACS patients suggests their potential role in modulating immune responses and disease progression.

ACS is a progressive condition characterized by the buildup of lipids within arteries. The pathogenesis of ACS is significantly influenced by disorders in lipid metabolism. Interestingly, MDSCs showed an inverse correlation with two crucial risk factors LDL and TC (*Wen et al., 2022*). It is necessary to investigate the associations between M-MDSCs levels and additional clinical characteristics of ACS patients. Firstly, this study demonstrated significantly higher M-MDSCs frequencies in patients with STEMI compared to NSTEMI and UA patients. This observation highlights the potential link between M-MDSCs and the pathophysiology of STEMI, which involves complete occlusion of the coronary arteries. Furthermore, this research established a correlation between M-MDSCs levels and the severity of coronary artery stenosis. Patients with more severe stenosis exhibited higher frequencies of M-MDSCs compared to those with milder stenosis. This result suggesting that MDSCs may contribute to the progression of coronary artery disease and the formation of more severe stenotic lesions. It supports the notion that the accumulation of M-MDSCs in peripheral blood reflects ongoing inflammation within the coronary arteries. The association with the number of coronary artery lesions further underscores the relationship between M-MDSCs and the extent of coronary artery involvement. Thirdly, patients with multi-vessel disease had significantly increased M-MDSCs frequencies compared to those with single-vessel disease. This finding is consistent with the hypothesis that a broader inflammatory response may lead to greater expansion of M-MDSCs. It emphasizes the importance of M-MDSCs as potential biomarkers for assessing disease severity and complexity. Fourthly, the association between M-MDSCs levels and cardiac function, as assessed by the Killip classification, revealed that patients with more severe cardiac dysfunction had higher frequencies of MDSCs. This finding suggests that M-MDSCs may be involved in the pathogenesis of myocardial injury and remodeling in ACS. The association of MDSCs with the severity of stenosis suggests their contribution to the pathogenesis of ACS and their potential as a therapeutic target.

Obesity is the most critical factor in the development of coronary heart disease (*Atique et al., 2016*). Interestingly, our study highlights a significant association between obesity and increased M-MDSCs levels in ACS patients. This observation provides new insights into the relationship between obesity and coronary artery disease, suggesting that M-MDSCs may play a role in modulating the effects of obesity-related chronic low-grade inflammation. While previous research has established the link between obesity and MDSCs. Molecular markers of obesity (*e.g.*, IL-6, CRP, IL-1 $\beta$) increase the generation of MDSCs (*Clements et al., 2018*). MDSCs exhibit a protective role by reducing both systemic inflammation and inflammation within adipose tissue. These MDSCs play a dual role in obesity-related conditions by providing protection against certain metabolic dysfunctions associated with obesity (*Clements et al., 2018*; *Xia et al., 2011*). Therefore, we hypothesize that the elevated levels of MDSCs in ACS patients also play a protective role by reducing inflammatory responses.

Consistent with similar results in diffuse large B-cell lymphoma and systemic lupus erythematosus patients, the level of M-MDSCs significantly decreased after therapy (*Wang et al., 2021*; *Wang et al., 2019*). This study found the level of M-MDSCs significantly decreased after PCI surgery and anticoagulant therapy. This observation suggests that the level of M-MDSCs correlates with disease progression. As the disease returns to a normal state, the frequency of M-MDSCs decreases significantly. The significant changes in M-MDSCs frequency after therapy in ACS patients can be used as an indicator of treatment efficacy.

The chronic, low-grade inflammation is the dominant driving force for MDSCs accumulation (*Ostrand-Rosenberg & Sinha, 2009*). The pro-inflammatory mediators IL-6, IL-10, GM-CSF, and TNF-$\alpha$ are major inducers of the differentiation, accumulation, and potency of MDSCs (*Bai et al., 2022*; *Rosborough et al., 2012*; *Wu et al., 2017*; *Zheng et al., 2021*). In terms of cytokine profiles, our research demonstrated significantly increased plasma concentrations of IL-6, IL-10, GM-CSF, and TNF-$\alpha$ in ACS patients compared to SAP patients and healthy controls. These cytokines are involved in inflammation and immune regulation. The elevated levels of these cytokines in ACS patients may contribute to the expansion and activation of M-MDSCs. The positive correlations between M-MDSCs levels and the expression of IL-6, GM-CSF further support the expansion role of cytokine in ACS. These findings align with previous research, indicating the association between cytokine dysregulation and MDSCs expansion in different disease contexts.

Activated MDSCs secrete cytokines and other soluble factors that exert immunoregulatory effects on various immune cells (*Bekic & Tomic, 2023*; *Jimenez-Cortegana & Galluzzi, 2023*). These cytokines, such as Arg-1, and iNOS by MDSCs, collectively contribute to the inhibition of T and NK cell proliferation. Regarding MDSC-related enzymes, our study showed a significant increase in Arg-1 mRNA levels in ACS patients compared to SAP patients and healthy controls. This finding is consistent with previous research (*Cao et al., 2022*), suggesting the involvement of Arg-1 in the immunosuppressive function of MDSCs. The positive correlations between M-MDSCs levels and the expression of Arg-1 further support the immunosuppressive role of M-MDSCs in ACS. Although the difference in iNOS mRNA levels did not reach statistical significance, there was a trend

towards increased iNOS expression in ACS patients. Further research is needed to fully understand the significance of iNOS in ACS and its interaction with M-MDSCs.

The logistic regression analysis revealed that diabetes, TG, HDL-C and M-MDSCs were significant risk factors associated with the severity of coronary artery stenosis in ACS patients. Among these factors, TG emerged as the most significant risk factor, indicating its potential role in promoting atherosclerosis and plaque instability. Interestingly, HDL-C exhibited a significant inverse association with the degree of coronary artery stenosis. This finding suggests that HDL-C are associated with a lower degree of coronary artery stenosis, indicating a potential protective role of HDL-C against the development and progression of atherosclerosis in the coronary arteries. Further ROC curve analysis suggested that TG, HDL-C, and M-MDSCs levels have varying diagnostic performances in predicting the severity of coronary artery stenosis. TG and HDL-C provide moderate to good performance, with TG showing higher specificity and HDL-C showing higher sensitivity. M-MDSCs levels demonstrate good performance with balanced sensitivity and specificity. Integrating these factors into a comprehensive risk assessment model may enhance the accuracy of predicting the severity of coronary artery stenosis and inform clinical decision-making.

## CONCLUSIONS

In conclusion, this research reveals a significant association between M-MDSCs levels and the severity of coronary artery stenosis in patients with ACS. The expansion of M-MDSCs suggests their potential role in modulating immune responses, contributing to the progression of coronary artery disease and an indicator of treatment efficacy. Furthermore, increased M-MDSCs indicating a potential role for modulating the effects of ACS-related inflammation. Additionally, ACS patients exhibited elevated levels of pro-inflammatory cytokines and expression of immunosuppressive factors by MDSCs, supporting their immunoregulatory and suppressive functions. Finally, M-MDSCs demonstrated a balanced sensitivity and specificity in predicting the severity of coronary artery stenosis. Integrating these factors into a comprehensive risk assessment model may enhance the accuracy of predicting the severity of coronary artery stenosis in ACS patients.

### Funding
This work was supported by grants from projects funded by An-hui Natural Fund General Project (2008085MH239). There was no additional external funding received for this study. The funders had no role in study design, data collection and analysis, decision to publish, or preparation of the manuscript.

### Grant Disclosures
The following grant information was disclosed by the authors:
An-hui Natural Fund General Project: 2008085MH239.

## Competing Interests

The authors declare there are no competing interests.

## Author Contributions

- Yinsheng Tan conceived and designed the experiments, prepared figures and/or tables, and approved the final draft.
- Mingde Ren conceived and designed the experiments, prepared figures and/or tables, and approved the final draft.
- Jie Hou performed the experiments, prepared figures and/or tables, authored or reviewed drafts of the article, and approved the final draft.
- Tao Hou analyzed the data, authored or reviewed drafts of the article, and approved the final draft.
- Xianhe Lin conceived and designed the experiments, authored or reviewed drafts of the article, and approved the final draft.

## Human Ethics

The following information was supplied relating to ethical approvals (i.e., approving body and any reference numbers):

The studies involving human participants were reviewed and approved by the Ethics Committee of the First Affiliated Hospital of Anhui Medical University (No. S20200021). The studies were conducted in accordance with the local legislation and institutional requirements. The participants provided their written informed consent to participate in this study.

## Data Availability

The data is available at Zenodo: Xianhe, L., Yinsheng, T., Minde, R., Tao, H., & Jie, H. (2023). Raw FCM Data of ACS SAP patients [Data set]. Zenodo. Available at https://doi.org/10.5281/zenodo.12074209.

## Supplemental Information

Supplemental information for this article can be found online at http://dx.doi.org/10.7717/peerj.18154#supplemental-information.

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
