# Peer review of "Increased CD14+HLA-DR-/low myeloid-derived suppressor cells can be regarded as a biomarker on disease severity and response to therapy in acute coronary syndrome"

_PeerJ, doi:10.7717/peerj.18154_

## Round 0.1 · original submission · Minor Revisions

· Academic Editor

Minor Revisions

The aim of the manuscript is scientifically good. The manuscript needs a minor revision. Please address the reviewers' comments and resubmit the corrected version.

Reviewer 1 ·

Basic reporting

1.The manuscript is written in clear and professional English. The language is unambiguous, and the scientific terminology is appropriately used. However, a thorough proofreading to correct minor grammatical errors and ensure clarity and conciseness is recommended.
2.The paper provides adequate background information and context, citing relevant literature to support the study's rationale and significance. The references are up-to-date and cover key studies in the field of cardiovascular immunology and myeloid-derived suppressor cells (M-MDSCs).
3.The manuscript follows a standard scientific article structure, including Abstract, Introduction, Methods, Results, Discussion, and References. Figures and tables are appropriately used to present data, though ensuring high resolution and clear labeling will enhance their effectiveness. The raw data has been shared, which supports transparency and reproducibility.
4.The manuscript is self-contained, presenting all necessary results related to the hypotheses. The findings are relevant and directly address the research question, providing a comprehensive understanding of the study's outcomes.

Experimental design

1.The research is original and fits well within the aims and scope of the journal. It adds valuable insights into the role of M-MDSCs in acute coronary syndrome (ACS), filling a significant knowledge gap in this area.
2.The research question is clearly defined, relevant, and meaningful. The study aims to investigate the dynamic changes in M-MDSCs and their implications in ACS pathogenesis, which is a significant area of interest in cardiovascular research.
3.The investigation is rigorous and adheres to high technical and ethical standards. The use of multi-parameter flow cytometry and appropriate patient cohorts (ACS, SAP, and healthy controls) strengthens the study's validity.
4.The methods are described in sufficient detail, allowing for replication of the study. However, providing more specific details about the flow cytometry protocols, including the antibodies used and gating strategies, would further enhance reproducibility.

Validity of the findings

1.While the review does not assess the impact and novelty, the study's findings are novel and provide meaningful contributions to the literature. The rationale for replication is clearly stated, highlighting the benefit to the field.
2.All underlying data have been provided, and the data analysis is robust and statistically sound. The study includes appropriate controls and statistical tests to validate the findings.
3.The conclusions are well stated and directly linked to the original research question. They are appropriately limited to the supporting results, providing a clear and concise summary of the study's implications.

Additional comments

1.the introduction could be improved by succinctly summarizing existing knowledge and more clearly defining the study's unique contributions to the field.
2.The methods section should include more information on how patients were diagnosed with ACS, SAP, and classified into various severity groups should be explicitly stated.
3.Provide more detailed descriptions of the flow cytometry protocols, including the specific antibodies used and gating strategies.

Reviewer 2 ·

Basic reporting

No comment.

Experimental design

No comment.

Validity of the findings

No comment.

Annotated reviews are not available for download in order to protect the identity of reviewers who chose to remain anonymous.

---

## Round 0.2 · accepted · Accept

· Academic Editor

Accept

Dear Dr. Lin,
Thank you for your submission to PeerJ.
I am writing to inform you that you replied to the reviewers comments very well and the resubmitted manuscript - Increased CD14+HLA-DR-/lowmyeloid-derived suppressor cells can be regarded as a biomarker on disease severity and response to therapy in acute coronary syndrome - has been Accepted for publication. Congratulations!

Reviewer 1 ·

Basic reporting

no comment

Experimental design

no comment

Validity of the findings

no comment

Additional comments

no comment

Reviewer 2 ·

Basic reporting

No comment

Experimental design

No comment

Validity of the findings

No comment